# Med-UniC: Unifying Cross-Lingual Medical Vision-Language Pre-Training by Diminishing Bias

**Zhongwei Wan**[1][*], **Che Liu**[2][*], **Mi Zhang**[1][†], **Jie Fu**[5][†], **Benyou Wang**[4], **Sibo Cheng**[2],
**Lei Ma**[3], **César Quilodrán-Casas**[2], **Rossella Arcucci**[2]

[1]The Ohio State University    [2]Imperial College London
[3]Peking University    [4]The Chinese University of Hong Kong, Shenzhen
[5]The Hong Kong University of Science and Technology

{wan.512, mizhang.1}@osu.edu, lei.ma@pku.edu.cn, wangbenyou@cuhk.edu.cn
{che.liu21, sibo.cheng, c.quilodran, r.arcucci}@imperial.ac.uk, jiefu@ust.hk

## Abstract

The scarcity of data presents a critical obstacle to the efficacy of medical vision-language pre-training (VLP). A potential solution lies in the combination of datasets from various language communities. Nevertheless, the main challenge stems from the complexity of integrating diverse syntax and semantics, language-specific medical terminology, and culture-specific implicit knowledge. Therefore, one crucial aspect to consider is the presence of community bias caused by different languages. This paper presents a novel framework named Unifying Cross-Lingual Medical Vision-Language Pre-Training (**Med-UniC**), designed to integrate multi-modal medical data from the two most prevalent languages, English and Spanish. Specifically, we propose **C**ross-lingual **T**ext Alignment **R**egularization (**CTR**) to explicitly unify cross-lingual semantic representations of medical reports originating from diverse language communities. **CTR** is optimized through latent language disentanglement, rendering our optimization objective to not depend on negative samples, thereby significantly mitigating the bias from determining positive-negative sample pairs within analogous medical reports. Furthermore, it ensures that the cross-lingual representation is not biased toward any specific language community. **Med-UniC** reaches superior performance across 5 medical image tasks and 10 datasets encompassing over 30 diseases, offering a versatile framework for unifying multi-modal medical data within diverse linguistic communities. The experimental outcomes highlight the presence of community bias in cross-lingual VLP. Reducing this bias enhances the performance not only in vision-language tasks but also in uni-modal visual tasks. The source code has been released at https://github.com/SUSTechBruce/Med-UniC.

## 1   Introduction

English, despite not being the primary native language for a vast majority of the global population, remains the dominant language in vision-language pre-training (VLP) datasets. Uni-lingual VLP models not only demonstrate restricted performance in cross-lingual tasks, but also bring the community bias on non-English speaking populations (displayed in Fig 1), particularly in the context of medical applications.

Researchers have used machine-translated non-English corpora and techniques like masked language model (MLM) and contrastive learning to unify cross-lingual representations [1–

---

[*] Equal Contribution.
[†] Corresponding Authors

37th Conference on Neural Information Processing Systems (NeurIPS 2023).

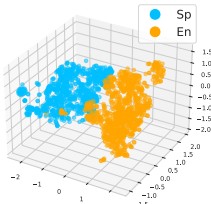 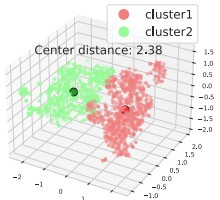 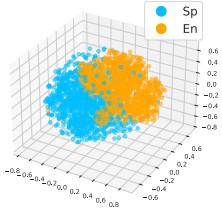 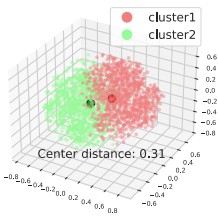

(a) 3D Text Embedding from MLM.

(b) 3D Clustering from MLM.

(c) 3D Text Embedding from Med-UniC.

(d) 3D Clustering from Med-UniC.

Figure 1: Graphical illustration of community bias in MLM and Med-UniC is shown with blue points representing Spanish reports and orange points indicating English reports, visualized via T-SNE. MLM denotes cross-lingual masked language modeling.

4]. However, MLM-based representations still separate languages, as shown in Fig 1a and 1b. Additionally, as highlighted in Fig 2, the significant similarity among reports from diverse communities suggests that the strict reliance on negative samples in contrastive learning could introduce more bias during text alignment. Hence, Med-UniC focuses on learning text invariants in VLP using negative-free text alignment to avoid the bias mentioned above.

In this work, we introduce a unified framework named **Med-UniC**, designed to acquire language-agnostic representations from chest x-ray (CXR) scans and associated radiology reports. Med-UniC learns the representation from 3 perspectives: visual invariants, visual-textual invariants, and text invariants. Considering medical vision-language tasks, such as zero-shot image classification, are dependent on semantic information and language-independent, we propose **CTR** (**C**ross-lingual **T**ext Alignment **R**egularization) to explicitly minimize linguistic disparities in cross-lingual representations within the latent space, as visualized in Fig 1c and 1d. Consequently, Med-UniC tackles non-English vision-language tasks without the model bias stemming from the language model (LM) pre-trained on predominantly English corpora. Additionally, we found that the unified cross-lingual representation enhances performance across a range of uni-modal visual tasks. This paper makes the following contributions:

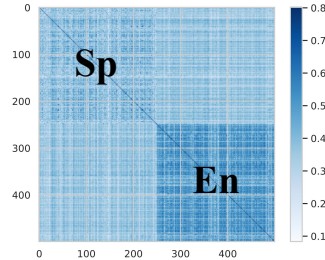

Figure 2: Similarity matrix for medical reports embedding from MLM.

- To the best of our knowledge, we are the first to empirically identify the existence of community bias originating from diverse languages in cross-lingual medical VLP (displayed in Fig 1), and the negative impact of community bias on both uni-modal and vision-language downstream tasks.

- We introduce the framework Med-UniC with CTR, designed to diminish community bias in medical VLP by unifying cross-lingual latent representations. Med-UniC achieves SOTA results in medical vision-language tasks across different languages, demonstrating its efficacy and broad applicability across various language communities.

- Med-UniC achieves SOTA results on all uni-modal visual tasks. This highlights the advantages of mitigating community bias and unifying cross-lingual representations in medical VLP, enabling robust and comprehensive learning of visual representations.

- Med-UniC effectively mitigates community bias without the requirement for manual curation or language-specific annotations. Importantly, Med-UniC enhances the accessibility of medical VLP to non-English speaking populations, circumventing potential biases that may arise from predominantly English datasets in traditional VLP.

## 2 Related Work

**Medical VLP** Complex medical reports and a shortage of large-scale medical image-text datasets have limited medical VLP research. Previous works such as ConVIRT [5] and GLoRIA [6] utilized contrastive objectives and global-local VLP to align image-text pairs. MGCA [7] used disease-level annotations for alignment, while MedKLIIP [8] manually extracted medically relevant entities.

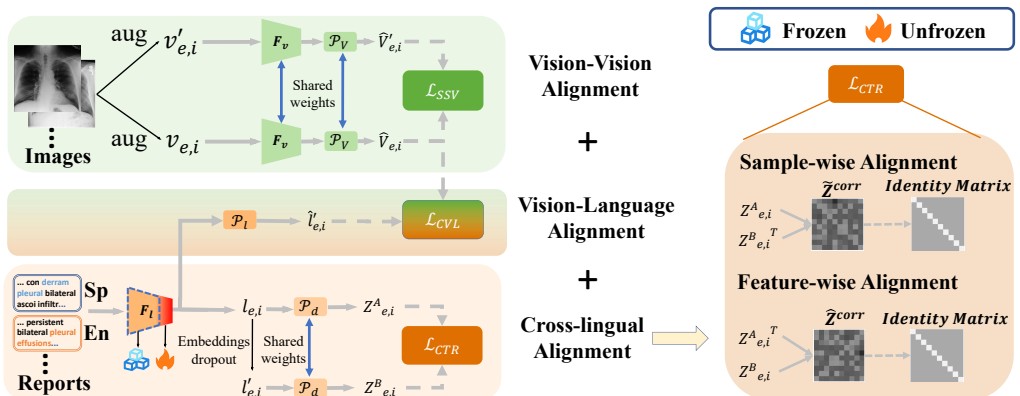

Figure 3: Overview Med-UniC. CVL, SSV, CTR represent *cross-lingual vision-language alignment*, *self-supervised vision alignment*, *Cross-lingual Text Alignment Regularization*, respectively.

MRM [9] replaced alignment with a reconstruction task involving masked visual and textual tokens. However, these methods fail to capitalize on valuable cross-lingual knowledge and do not effectively unify cross-lingual representations. As a result, their performance on cross-lingual tasks is considerably limited.

**Cross-lingual VLP**   Most recent cross-lingual VLP research has focused on a limited set of languages, with a few comparing natural cross-lingual VLP to English-only VLP [10]. Studies such as [1–4] used machine-translated corpora and contrastive learning to generate language-agnostic representations. Yet, cross-lingual VLP in the medical domain remains unexplored mainly due to two main challenges: the limitations of machine translation for medical reports and the bias introduced by hard positive-negative pairs in contrastive learning. We propose CTR, a negative-free disentangled loss method, to unify cross-lingual text representations, addressing these issues.

**Large General Model**   Recently, there have been impressive advances in large general models for language and vision tasks, such as ChatGPT [11], SAM, and DINOv2 [11–13]. However, these models still face significant limitations in the medical domain due to their lack of domain-specific knowledge and inability to jointly process visual and textual information [14–16]. ChatGPT [11] and SAM [12], currently limited to single-modality input, are unsuitable for vision-language tasks. While SAM excelled at instance segmentation, it struggled with medical image pathology segmentation [16–18]. Conversely, Med-UniC addressed these constraints by utilizing visual and textual data, integrating domain-specific knowledge without the high computational expenses of large general models. Surprisingly, Med-UniC outperformed the extensive vision model, DINOv2 [13], trained on 1.2B images with 1B parameters, by a considerable margin in multiple visual tasks. This suggests that Med-UniC is a more effective and efficient solution compared to large general models.

## 3   Method

### 3.1   Overall Framework of Med-UniC

Our Med-UniC framework aims to learn cross-lingual medical representation from CXR images and paired radiology reports. Given a training set of $N$ cross-lingual dataset $S \in \mathcal{V} \times \mathcal{L}$ consisting of pairs $(v_{e,i}, l_{e,i})$, where $\mathcal{V}$ and $\mathcal{L}$ are visual and text set, respectively, $v_{e,i}$ is a raw image and $l_{e,i}$ is a text report, $e$ belongs to language domain (e.g., Spanish or English), $i \in N$. The Med-UniC architecture mainly consists of an image encoder $\mathcal{F}_v : \mathcal{V} \mapsto \mathbb{R}^{D_v}$ to encoder the raw image into the embeddings with dimension $D_v$, and a cross-lingual text encoder $\mathcal{F}_l : \mathcal{L} \mapsto \mathbb{R}^{D_l}$ to encoder the text report to the embeddings with dimension $D_l$. Then $\mathbf{S} = \{(\mathbf{v}_{e,1}, \mathbf{l}_{e,1}), (\mathbf{v}_{e,2}, \mathbf{l}_{e,2}), \ldots, (\mathbf{v}_{e,N}, \mathbf{l}_{e,N})\}$, where $\mathbf{v}_{e,i} = \mathcal{F}_v(v_{e,i})$ and $\mathbf{l}_{e,i} = \mathcal{F}_l(l_{e,i})$.

As depicted in Fig 3, Med-UniC incorporates three concurrent alignment strategies: *cross-lingual vision-language alignment*, *self-supervised vision alignment*, and *cross-lingual text alignment regularization*. For cross-lingual vision-language alignment, we extend the text encoder from uni-lingual into a cross-lingual style, which can equip the model with cross-lingual cross-modal ability by pulling the embeddings of paired image-reports pairs together while pushing apart embeddings of unmatched pairs, under the loss $\mathcal{L}_{CVL}$. Meanwhile, we employ self-supervised vision alignment, leveraging the

loss $\mathcal{L}_{SSV}$, to enhance the robustness of visual representation [19]. More importantly, we introduce a cross-lingual text alignment regularization, encompassing sample-level and feature-level approaches, to mitigate community bias stemming from different languages. This regularization is supervised by the loss $\mathcal{L}_{CTR}$. The learning objective of Med-UniC integrates the three strategies mentioned above and can be formulated as follows:

$$\mathcal{L} = \mathcal{L}_{\text{CVL}} + \mathcal{L}_{\text{SSV}} + \mathcal{L}_{\text{CTR}} \qquad (1)$$

This training scheme compels Med-UniC to assimilate information from diverse perspectives, fostering a robust cross-modal, cross-lingual framework that concurrently learns visual invariants, visual-textual invariants, and text invariants.

## 3.2 Cross-lingual Vision-language Alignment

**Cross-lingual Medical LM** To initialize Med-UniC with the ability to process different languages and learn fundamental cross-lingual syntactic and semantic knowledge. We select CXR-BERT [20], a uni-lingual LM pre-trained on a large-scale biomedical corpus, as our text encoder and further adapted it for cross-lingual operation. The concrete steps proceed as follows: **(1)** Constructing a cross-lingual vocabulary set $\mathcal{T}$: we collected the radiology reports of the second language (e.g., Spanish PadChest dataset [21]), which is seldom used for medical pre-training compared to English dataset [22, 23]. Then we leverage a general Spanish Spacy [3] to build a tokenizer and make use of TF-IDF tool [4] to generate $M$ ranked new tokens $\mathcal{T}_{sp} = \{t_{sp}^1, t_{sp}^1, \ldots, t_{sp}^M\}$ according to their importance from multiple reports. **(2)** Building new wording embeddings $\mathbf{W}$: we augment the original vocabulary $\mathcal{T}_{en}$ with the sorted Spanish tokens to generate $\mathcal{T} = \{\mathcal{T}_{en}, \mathcal{T}_{sp}\}$ then expand a length of $M$ random initialized vectors $\mathbf{W}_{sp}$ on CXR-BERT's word embeddings $\mathbf{W}_{en}$, where $\mathbf{W} = [\mathbf{W}_{en}; \mathbf{W}_{sp}]$. **(3)** Masked Cross-lingual Modeling: we following BERT [24], randomly mixed English [22] and Spanish [21] medical reports as pre-train corpus. Then we randomly choose tokens in the mixed sequences, replace them with the [MASK] token and set 15% masking probability as in BERT [25]. We selectively update several high layers to alleviate catastrophic forgetting [26] during vision-language pre-training. More details will be show in Sec 4.5.

**Vision-Language Alignment** Following CLIP framework [27], we incorporate a contrastive learning object to predict the matched pair $(v_{e,i}, l_{e,i})$ from $N \times N$ possible image-text pairs while mapping $N^2 - N$ negative pairs far apart. Specifically, two non-linear visual and linguistic projectors $\mathcal{P}_l$ and $\mathcal{P}_v$ are used to transform $\mathbf{v}_{e,i}$ and $\mathbf{l}_{e,i}$ into the same dimension $d$, where $\hat{\mathbf{v}}_{e,i} = \mathcal{P}_v(\mathbf{v}_{e,i}), \hat{\mathbf{l}}_{e,i} = \mathcal{P}_l(\mathbf{l}_{e,i})$, and $\hat{\mathbf{v}}_{e,i}, \hat{\mathbf{l}}_{e,i} \in \mathbb{R}^d$. Obtaining image feature vectors $[\hat{\mathbf{v}}_{e,i}]_{i=1}^N$ and text feature vectors $[\hat{\mathbf{l}}_{e,i}]_{i=1}^N$ from a training batch, we compute cosine similarities $s_{i,i}^{v2l} = \hat{\mathbf{v}}_{e,i}^\top \hat{\mathbf{l}}_{e,i}$ and $s_{i,i}^{l2v} = \hat{\mathbf{l}}_{e,i}^\top \hat{\mathbf{v}}_{e,i}$, representing image-text and text-image similarities, respectively. $\mathcal{L}_{\text{CVL}}$ is then formulated as follows:

$$\mathcal{L}_v^{v2l} = -\log \frac{\exp(s_{i,i}^{v2l}/\sigma_1)}{\sum_{j=1}^K \exp(s_{i,j}^{v2l}/\sigma_1)}, \mathcal{L}_i^{l2v} = -\log \frac{\exp(s_{i,i}^{l2v}/\sigma_1)}{\sum_{j=1}^K \exp(s_{i,j}^{l2v}/\sigma_1)} \qquad (2)$$

$$\mathcal{L}_{\text{CVL}} = \frac{1}{2K} \sum_{i=1}^N \left( \mathcal{L}_v^{v2l} + \mathcal{L}_l^{l2v} \right), \qquad (3)$$

where $\mathcal{L}_v^{v2l}$ and $\mathcal{L}_l^{l2v}$ are image-text and text-image InforNCE [28] contrastive loss, respectively. $\sigma_1$ is the temperature hyper-parameter set to 0.07 in our experiments, $K$ is the batch size for each step and $K \in N$. Through overall loss $\mathcal{L}_{\text{CVL}}$, the model learns maximal mutual information between the matched image-text pairs containing cross-lingual attributes within a batch.

## 3.3 Self-supervised Vision Alignment

To obtain more exhaustive visual representation, we include visual invariant learning as a parallel objective during VLP. Drawing inspiration from [19], we initially apply random augmentations (such as random cropping and flipping) to the original images to create augmented views as positive pairs $[(v_{e,i}, v'_{e,i})]_{i=1}^N$, while treating the rest of the images in the mini-batch as negative samples. We follow the data augmentation strategy as per [19]. Subsequently, we extract the representations of

---

[3]https://spacy.io/models/es-dep-news-trf
[4]TfidfVectorizer: https://scikit-learn.org/

the augmented views $[\hat{\mathbf{v}}']_{i=1}^N$ using the vision projector $p_v$ and vision encoder $\mathcal{F}_v$, similar to $[\hat{\mathbf{v}}]_{i=1}^N$. Therefore, the visual invariant learning objective becomes:

$$\mathcal{L}_{SSV} = -\frac{1}{K}\sum_{j=1}^N \log \frac{\exp(s_{i,i}^{v2v'}/\sigma_2)}{\sum_{j=1}^N \exp(s_{i,j}^{v2v'}/\sigma_2)}, s_{i,i}^{v2v'} = \hat{\mathbf{v}}_{e,i}^\top \hat{\mathbf{v}}_{e,i}' \tag{4}$$

where $\sigma_2$ is the temperature hyper-parameter also set to 0.07 for overall loss objective $\mathcal{L}_{SSV}$.

### 3.4 Cross-lingual Text Alignment Regularization

As illustrated in Fig 1, and corroborated by research in natural language processing [29, 30], cross-lingual text representations tend to form distinct clusters based on their respective languages. This trend introduces a community bias within data from different language communities, even when no explicit language attribute is provided. This suggests that VLP processes medical data based on their language community, risking unfairness in clinical applications and potentially decreasing downstream task performance in both uni-modal and vision-language tasks [31–33]. To mitigate this bias and potential risks, we introduce *Cross-lingual Text Alignment Regularization* (CTR) to learn language-independent text representations and neutralize the adverse effects of community bias on other modalities. CTR comprises three components:

**Text augmentation** We first adopt the dropout strategy [34] to generate augmented the text representation $l_{e,i}'$ from the text encoder and obtain the matched pairs $[(l_{e,i}, l_{e,i}')]_{i=1}^N$, and then a separate linguistic projector $\mathcal{P}_d$ designed for de-correlation is leveraged to generate two different view pairs $[(\mathbf{Z^A}_{e,i}, \mathbf{Z^B}_{e,i})]_{i=1}^N$, where $\mathbf{Z^A}_{e,i} = \mathcal{P}_d(l_{e,i})$, $\mathbf{Z^B}_{e,i} = \mathcal{P}_d(l_{e,i}')$, and the new feature dimension $D' > D$.

**Text-feature alignment** To further alleviate information redundancy [35, 36] and obtain the shared cross-lingual text representation, we first normalize the augmented embedding pairs $\{\mathbf{Z_e^A}, \mathbf{Z_e^B}\} \in \mathbb{R}^{N \times D'}$ along the batch $K$ dimension so that each feature dimension has a zero-mean and $1/\sqrt{K}$ standard deviation distribution to generate $\tilde{\mathbf{Z}}_e$, and then compute their cross-correlation $\tilde{\mathbf{Z}}_e^{corr} = \tilde{\mathbf{Z}}_e^{\mathbf{AT}}\tilde{\mathbf{Z}}_e^{\mathbf{B}}$. The formulas of feature-dimension decorrelation can be defined as:

$$\mathcal{L}_{TF} = \frac{1}{D'}\left\{ \underbrace{\sum_j^{D'}\left(1 - \sum_i^K \tilde{\mathbf{Z}}_{e,i}^{A,j\mathbf{T}}\tilde{\mathbf{Z}}_{e,i}^{B,j}\right)^2}_{\text{cross-lingual invariance}} + \lambda \underbrace{\sum_j^{D'}\sum_{i \neq j}^K \tilde{\mathbf{Z}}_{e,i}^{A,j\mathbf{T}}\tilde{\mathbf{Z}}_{e,i}^{B,j}}_{\text{cross-lingual gap reduction}} \right\}, \quad \tilde{\mathbf{Z}}_e = \frac{\mathbf{Z_e} - \mu_K(\mathbf{Z_e})}{\sqrt{K}\sigma(\mathbf{Z_e})} \tag{5}$$

The first term's objective is to learn a language-invariant representation by optimizing the diagonal elements of the correlation matrix to equal one. Simultaneously, the second term aims to shrink the cross-lingual gap and optimize information utilization in each latent dimension by driving the off-diagonal elements towards zero. Finally, We normalize the loss along with the feature dimension $D'$.

**Text-to-text alignment**: Similarly, the text-to-text alignment decorrelates the cross-correlation matrix along with feature dimension $D'$, and $\hat{\mathbf{Z}}_e$ is the normalized embeddings, $\hat{\mathbf{Z}}_e^{corr} = \hat{\mathbf{Z}}_e^{\mathbf{A}}\hat{\mathbf{Z}}_e^{\mathbf{BT}}$ :

$$\mathcal{L}_{TT} = \frac{1}{K}\left\{ \underbrace{\sum_j^K\left(1 - \sum_i^{D'} \hat{\mathbf{Z}}_{e,i}^{A,j}\hat{\mathbf{Z}}_{e,i}^{B,j,\mathbf{T}}\right)^2}_{\text{text instance alignment}} + \lambda \underbrace{\sum_j^K\sum_{i \neq j}^{D'} \hat{\mathbf{Z}}_{e,i}^{A,j}\hat{\mathbf{Z}}_{e,i}^{B,j,\mathbf{T}}}_{\text{text consistency regularizer}} \right\}, \quad \hat{\mathbf{Z}}_e = \frac{\mathbf{Z_e} - \mu_{D'}(\mathbf{Z}_e)}{\sqrt{D'}\sigma(\mathbf{Z_e})} \tag{6}$$

where the *text instance alignment* term attempts to maximize the mutual information of a batch of cross-lingual text samples, and the *text consistency regularizer* can also be deemed as the text in-modal consistency [37] by reducing the mismatched text pairs into 0 in a batch $K$. Where $\lambda$ in Eq 5, 6, is a non-negative hyperparameter trading off two terms. We also normalize the loss with the batch dimension $K$. Therefore, the loss of *Cross-lingual Text Alignment Regularization* $\mathcal{L}_{CTR}$ is:

$$\mathcal{L}_{CTR} = \mathcal{L}_{TF} + \mathcal{L}_{TT} \tag{7}$$

# 4 Experiments

## 4.1 Pre-training Configuration

**Dataset** We pre-train Med-UniC framework using MIMIC-CXR [38], which contains CXR images and their corresponding radiology reports in English. Also, we involve PadChest [39], which includes CXR images and their corresponding radiology reports collected in Valencia region, Spain. Both datasets are pre-processed following the approach described in [5–7], including image resizing, pixel value normalization, and text tokenization. Additionally, the dataset is filtered by excluding lateral views and reports with less than three tokens. This results in a pre-training dataset of approximately $220k$ image-text pairs for MIMIC-CXR [38] and $160k$ pairs for PadChest [39].

**Implementation** In the VLP stage, we employ ResNet-50 [40] and ViT [41] as the vision backbones. We report the linear classification results for these two vision encoders to illustrate the model-agnostic capabilities of Med-UniC. Med-UniC is trained over 50 epochs using an early stop strategy on 16 V100 GPUs with a batch size of 128 per GPU. We utilize AdamW [42] as the optimizer, setting the learning rate to $4e^{-5}$ and the weight decay to $5e^{-2}$. A linear warm-up and cosine annealing scheduler are also deployed in this process. Additionally, The coefficients $\lambda$ is set to $5.1e^{-3}$ following [36].

## 4.2 Downstream Tasks

**Medical Image Linear Classification** We perform this task on CheXpert [23], RSNA [43], and COVIDx [44] datasets. Following the previous work [5–7], we only update the parameter of a random initialized linear layer for classification and freeze the pre-trained vision backbone. We report the AUC scores (AUC) on CheXpert and RSNA and accuracy (ACC) on COVIDx as the evaluation metric following [6, 7].

**Medical Image Zero-shot Classification** We conduct this experiment on the CXP500 [45] and PDC [39] datasets, which comprise CXR images annotated by clinicians from English-speaking and Spanish-speaking countries, respectively. To circumvent prompt bias, we designate English positive prompt as '{*disease*}' and negative prompt as 'No {*disease*}'. Prompts in Spanish are prepared by native Spanish speakers, with the disease indicated as '{*disease*}' and the negative prompt represented as 'No hay {*disease*}'. Med-UniC is evaluated using both English and Spanish prompts across the two datasets, with additional experimental details provided in the Appendix. The results are represented as the macro average of AUC and F1 scores across all categories.

**Medical Image Semantic Segmentation** This task is performed on the RSNA [43] and the SIIM [46] datasets, following the data preprocessing in [6, 7]. Identical to [6, 7], the U-Net [47] fine-tuning settings are adopted for segmentation. All pre-trained vision backbones are considered as frozen encoders, and only the decoders of U-Net are updated during the fine-tuning. The segmentation performance is evaluated using Dice scores (Dice).

**Medical Image Object Detection** This task is performed on the RSNA [43] and Object-CXR [48] datasets, following the same preprocessing of [7]. Same as [7], we utilize YOLOv3 [49] as the detection architecture, using our pre-trained vision encoder as the backbone and only updating the detection head during fine-tuning. Mean Average Precision (mAP) with IOU thresholds 0.4∼0.75, is adopted to evaluate the detection task.

For all downstream tasks, except zero-shot classification, we fine-tune with $1\%, 10\%, 100\%$ of the training data. More downstream tasks' settings, including split information and train/valid/test set details, can be found in the Appendix.

## 4.3 Comparison to the state-of-the-art

**Zero-shot Classification** To assess the cross-lingual visual-textual representation learned from Med-UniC, we implement the zero-shot image classification task on two CXR datasets, which originate from distinct linguistic communities and utilize different language prompts. Tab 1 illustrates that Med-UniC surpasses all SOTA methods on both datasets, regardless of the language setting or the linguistic community data source. Across both datasets, Med-UniC delivers an average increase of over 20% in the F1 score when using English prompts and more than 15% when using Spanish

Table 1: Zero-shot Image Classification results. F1 and AUC scores are reported. Best results of each setting are in boldface. 'En' and 'Sp' respectively stand for prompts in English and Spanish languages. Methods with ⋆ leverage disease-level annotations for pre-training.

| Method | CXP500(En) AUC | F1 | CXP500(Sp) AUC | F1 | PDC(En) AUC | F1 | PDC(Sp) AUC | F1 |
|---|---|---|---|---|---|---|---|---|
| ConVIRT[5] | 59.5 | 19.2 | 60.5 | 15.8 | 45.1 | 26.5 | 49.1 | 12.6 |
| GLoRIA[6] | 43.2 | 2.4 | 40.2 | 16.1 | 52.3 | 10.1 | 50.3 | 8.2 |
| GLoRIA-MIMIC [6] | 46.2 | 5.5 | 51.5 | 20.3 | 53.1 | 12.1 | 52.2 | 11.3 |
| MGCA⋆ [7] | 72.1 | 6.5 | 50.4 | 22.3 | 46.4 | 32.5 | 49.8 | 26.1 |
| MedKILP⋆ [8] | 70.5 | 14.7 | 55.6 | 21.9 | 50.5 | 28.7 | 51.7 | 25.8 |
| MRM [9] | 65.2 | 10.4 | 48.3 | 16.1 | 50.1 | 24.6 | 51.4 | 25.1 |
| **Ours** | **75.4** | **30.3** | **71.3** | **32.2** | **72.9** | **42.6** | **71.4** | **37.1** |

prompts. This showcases the effectiveness and adaptability of Med-UniC in managing cross-lingual vision-language tasks.

Interestingly, the AUC score of other SOTA methods experiences a substantial drop when the prompts transition from English to Spanish on CXP500 [45], a dataset collected from English-speaking communities. Similarly, all compared methods show comparably poor performance on PDC [39], a dataset derived from Spanish-speaking communities. MedKLIP [8], despite its commendable performance on the CXP500 [45] in the English setting due to its supervised pre-training with disease annotation, persistently shows a drop in performance on both the CXP500 [45] and PDC [39] when Spanish prompts are used, and also on the PDC [39] when using English prompts. These results highlight the significant community bias inherent in uni-lingual medical VLP models, even those utilizing annotations for pre-training. This bias adversely impacts the diagnostic quality when dealing with patients from non-English-speaking communities or those who do not speak English.

The unsatisfied performance of the compared methods also suggests that these models might incorporate linguistic community attributes during VLP, which negatively influences the learning of semantic meanings. Consequently, As a result, these models have difficulties in effectively interpreting CXR scans or prompts from non-English communities, which substantially limits the models' transferability. Further analysis can be found in Sec 4.4.

Table 2: Linear classification results for CheXpert, RSNA, and COVIDx datasets with 1%, 10%, and 100% training data. The best results are highlighted in bold. A standard ResNet-50 backbone is denoted as *CNN-based*. Methods with ⋆ leverage disease-level annotations for pre-training.

| Method | CheXpert (AUC) 1% | 10% | 100% | RSNA (AUC) 1% | 10% | 100% | COVIDx (ACC) 1% | 10% | 100% |
|---|---|---|---|---|---|---|---|---|---|
| Random Init | 56.1 | 62.6 | 65.7 | 58.9 | 69.4 | 74.1 | 50.5 | 60.3 | 70.0 |
| ImageNet Init | 74.4 | 79.7 | 81.4 | 74.9 | 74.5 | 76.3 | 64.8 | 78.8 | 86.3 |
| *CNN-based* | | | | | | | | | |
| GLoRIA [6] | 86.6 | 87.8 | 88.1 | 86.1 | 88.0 | 88.6 | 67.3 | 77.8 | 89.0 |
| ConVIRT [5] | 85.9 | 86.8 | 87.3 | 77.4 | 80.1 | 81.3 | 72.5 | 82.5 | 92.0 |
| GLoRIA-MIMIC [6] | 87.1 | 88.7 | 88.0 | 87.0 | 89.4 | 90.2 | 66.5 | 80.5 | 88.8 |
| MedKLIP⋆ [8] | 86.2 | 86.5 | 87.7 | 87.3 | 88.0 | 89.3 | 74.5 | 85.2 | 90.3 |
| MGCA⋆ [7] | 87.6 | 88.0 | 88.2 | 88.6 | 89.1 | 89.9 | 72.0 | 83.5 | 90.5 |
| **Med-UniC (ResNet-50)** | **88.2** | **89.2** | **89.5** | **89.1** | **90.4** | **90.8** | **76.5** | **89.0** | **92.8** |
| *ViT-based* | | | | | | | | | |
| MRM [9] | 88.5 | 88.5 | 88.7 | 91.3 | 92.7 | 93.3 | 66.9 | 79.3 | 90.8 |
| MGCA⋆ (ViT-B/16) [7] | 88.8 | 89.1 | 89.7 | 89.1 | 89.9 | 90.8 | 74.8 | 84.8 | 92.3 |
| **Med-UniC (ViT-B/16)** | 89.4 | 89.7 | 90.8 | 91.9 | 93.1 | 93.7 | 80.3 | 89.5 | 94.5 |
| **Med-UniC (ViT-L/32)** | **89.9** | **90.5** | **91.2** | **92.2** | **93.8** | **94.5** | **81.5** | **91.8** | **95.2** |

**Medical Image Linear Classification**  To assess the quality of the visual representations learned by Med-UniC, we employ linear classification [50] on CheXpert [23], RSNA [43], and COVIDx [44]. As illustrated in Tab 2, Med-UniC framework consistently surpasses all uni-lingual pre-trained baselines across various settings and vision backbones. Significantly, even when MedKLIP [8] employs supervised VLP with disease-level annotation, Med-UniC consistently surpasses MedKLIP [8] across all tasks and settings. This exemplifies the adaptability and efficacy of the visual representations

Table 3: Results of semantic segmentation on SIIM and RSNA datasets and object detection on RSNA and Object-CXR datasets. The best results for each setting are highlighted in bold, and the '-' denotes mAP values smaller than 1%. Methods with ⋆ leverage disease-level annotations.

| | Semantic Segmentation (Dice) | | | | | | Object Detection (mAP) | | | | | |
| | SIIM | | | RSNA | | | RSNA | | | Object CXR | | |
| Method | 1% | 10% | 100% | 1% | 10% | 100% | 1% | 10% | 100% | 1% | 10% | 100% |
|---|---|---|---|---|---|---|---|---|---|---|---|---|
| Random | 9.0 | 28.6 | 54.3 | 6.9 | 10.6 | 18.5 | 1.0 | 4.0 | 8.9 | - | 0.5 | 4.4 |
| ImageNet | 10.2 | 35.5 | 63.5 | 34.8 | 39.9 | 64.0 | 3.6 | 8.0 | 15.7 | - | 2.9 | 8.3 |
| ConVIRT[5] | 25.0 | 43.2 | 59.9 | 55.0 | 67.4 | 67.5 | 8.2 | 15.6 | 17.9 | - | 8.6 | 15.9 |
| GLoRA[6] | 35.8 | 46.9 | 63.4 | 59.3 | 67.5 | 67.8 | 9.8 | 14.8 | 18.8 | - | 10.6 | 15.6 |
| GLoRIA-MIMIC [6] | 37.4 | 57.1 | 64.0 | 60.3 | 68.7 | 68.3 | 11.6 | 16.1 | 24.8 | - | 8.90 | 16.6 |
| MGCA⋆ [7] | 49.7 | 59.3 | 64.2 | 63.0 | 68.3 | 69.8 | 12.9 | 16.8 | 24.9 | - | 12.1 | 19.2 |
| MedKLIP⋆ [8] | 50.2 | 60.8 | 63.9 | 66.2 | 69.4 | 71.9 | 8.9 | 16.3 | 24.5 | - | 7.1 | 11.6 |
| **Ours** | **56.7** | **62.2** | **64.4** | **72.6** | **74.4** | **76.7** | **16.6** | **22.3** | **31.1** | **6.6** | **13.3** | **21.6** |

Table 4: Ablation study of Med-UniC on linear classification, semantic segmentation and zero-shot classification. The best results of each setting are in boldface.

| Learning Objective | | | | COVIDx (ACC) | | | RSNA (AUC) | | | SIIM (Dice) | | | CXP500 (F1) | | PDC (F1) | |
| SSV | CVL | CTR | MLM | 1% | 10% | 100% | 1% | 10% | 100% | 1% | 10% | 100% | En | Sp | En | Sp |
|---|---|---|---|---|---|---|---|---|---|---|---|---|---|---|---|---|
| | ✓ | | ✓ | 72.8 | 85.5 | 91.8 | 87.7 | 88.5 | 89.4 | 51.9 | 56.5 | 58.7 | 63.5 | 59.8 | 62.2 | 58.5 |
| ✓ | ✓ | | ✓ | 74.5 | 85.8 | 92.2 | 88.1 | 89.3 | 89.9 | 53.4 | 58.7 | 60.1 | 68.5 | 62.1 | 64.6 | 61.7 |
| ✓ | ✓ | ✓ | | 75.0 | 84.3 | 92.5 | 88.2 | 89.6 | 89.7 | 53.8 | 59.6 | 61.5 | 70.3 | 65.9 | 65.4 | 63.7 |
| ✓ | ✓ | ✓ | ✓ | **76.5** | **89.0** | **92.8** | **89.1** | **90.4** | **90.8** | **56.7** | **62.2** | **64.4** | **75.4** | **71.3** | **72.9** | **71.4** |

cultivated by Med-UniC. Furthermore, this implies that unifying cross-lingual text representations via CTR can also improve the performance of uni-modal visual tasks.

**Medical Image Semantic Segmentation and Object Detection** In Tab 3, we further assessed the representation acquired by Med-UniC on segmentation and detection tasks. Impressively, Med-UniC outperforms all SOTA methods across every data fraction in all four tasks. Notably, Med-UniC achieves a Dice score of 72.6% on RSNA segmentation with only 1% data fine-tuning, exceeding the performance of all other SOTA methods fine-tuned on 100% data. Furthermore, Med-UniC reaches a 6.6% mAP on the Object-CXR dataset using just 1% data fine-tuning, surpassing other methods that barely achieve a 1% mAP. These findings further highlight the advantageous effects of unifying cross-lingual representations on vision-language tasks and uni-modal visual tasks.

### 4.4 Ablation Study and Bias Analysis

In this section, we ablate components of Med-UniC and present their performance on linear classification, zero-shot image classification, and semantic segmentation in Table 4. In all tasks, the combinations of all learning objectives achieve the highest performance, highlighting each component's crucial role in Med-UniC. Med-UniC, when integrated with CTR, significantly outperforms the version with MLM in zero-shot tasks and nearly all uni-modal visual tasks.

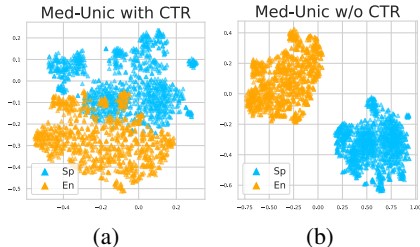

(a)          (b)

Figure 4: Visualization of image embeddings with or without CTR.

To further investigate the influence of CTR and MLM, we randomly select 1000 English and 1000 Spanish image-text pairs. We then illustrate the text and image embeddings in Fig 1 and Fig 4, respectively. Surprisingly, when the text encoder is pre-trained with MLM, we notice that the medical report embeddings tend to cluster by their respective languages, as shown in Fig 1a,1b. However, when employing CTR, the embeddings from diverse language reports draw nearer, leading to a reduction in the distance between the two clusters compared to the embeddings produced by MLM, as shown in Fig 1c,1d. This clear pattern illustrates CTR's ability to unify cross-lingual text representation within the latent space. Intriguingly, when pre-trained with CTR, the image embeddings become less distinguishable by their language community. In contrast, they form separate clusters according to their language community when pre-training does not involve CTR. This observation implies that community bias affects not only the text modality but also the

visual modality, as shown in Fig 4. Consequently, the reduction in community bias contributes to superior performance across all tasks when Med-UniC is pre-trained with all learning objectives. More details can be found in the Appendix.

## 4.5 Further Analysis

**Visualization of Language-agnostic Visual-textual Attention** We utilize Grad-CAM [51] to illustrate the areas in CXR images corresponding to various disease terminologies in both English and Spanish, as shown in Fig 5. Evidently, Med-UniC can accurately capture relevant regions that align closely with the indicated disease, exhibiting robustness across various languages. Consequently, the consistent cross-lingual visual-textual representations cultivated by Med-UniC underscore its outstanding generalizability across multiple downstream tasks and language communities.

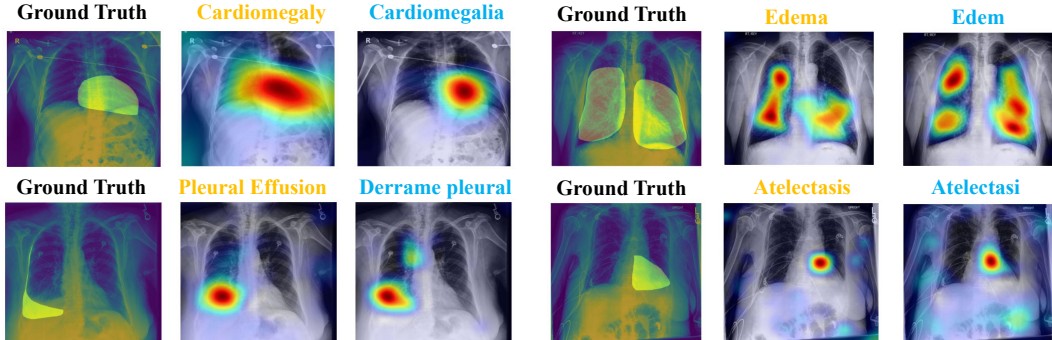

Figure 5: Attention heat-maps of the visual-textual association learned by Med-UniC, compared with ground truth annotations provided by clinicians. The blue and orange words denote Spanish and English types, respectively.

Table 5: Comparison with large vision model.

| Method | CheXpert 10% | CheXpert 100% | RSNA 10% | RSNA 100% | COVIDx 10% | COVIDx 100% |
|---|---|---|---|---|---|---|
| DINOv2 [52] | 81.6 | 83.2 | 84.5 | 86.3 | 85.0 | 92.2 |
| **Med-UniC(ViT-B/16)** | 89.7 | 90.8 | 93.1 | 93.7 | 89.5 | 94.5 |
| **Med-UniC(ViT-L/32)** | **90.5** | **91.2** | **93.8** | **94.5** | **91.8** | **95.2** |

Table 6: Results of Med-UniC with different numbers of frozen layers.

| Frozen layers | COVIDx (AUC) 10% | COVIDx (AUC) 100% | RSNA (Dice) 10% | RSNA (Dice) 100% |
|---|---|---|---|---|
| 0 | 87.7 | 92.5 | **75.0** | 75.8 |
| 6 | 88.0 | 92.3 | 74.3 | 76.3 |
| 9 | **89.0** | **92.8** | 74.4 | **76.7** |
| 12 | 87.1 | 90.5 | 72.3 | 74.4 |

**Comparison with Large Vision Model** In a comparison with DINOv2 [52], a large vision model trained on 1.2 billion images and comprising 1 billion parameters, Med-UniC outperforms it in linear classification tasks across all datasets, using different data ratios and two ViT backbones, as shown in Tab 5. This demonstrates the remarkable effectiveness of Med-UniC, even in scenarios with limited domain-specific data.

**Impact of Frozen Layers for LM** To investigate the impact of freezing layers in LM, we experimented with freezing various numbers (0, 6, 9, 12) in a 12-layer LM. Tab 6 shows that updating the last three layers offers better results comparable to updating more layers, hinting that updating more might cause catastrophic forgetting [26, 53] of cross-lingual MLM-acquired semantics. Performance declined when all layers were frozen, indicating the necessity of properly updating layers.

**Error Bars** We conducted three separate runs of Med-UniC, using different random seeds and ResNet-50 as the vision backbone for three tasks. We then reported the average metric and its standard deviation. As indicated in Tab 7, the variability between different task runs is relatively minor, demonstrating that Med-UniC consistently performs well in a variety of experiment configurations.

Table 7: Error bars analysis of linear classification, semantic segmentation, and object detection .

| Ratio | COVIDx(ACC) | RSNA(Dice) | Object CXR(mAP) |
|---|---|---|---|
| 1% | 76.54±0.32 | 72.60±0.33 | 6.62±0.67 |
| 10% | 89.01±0.16 | 74.43±0.19 | 13.34±0.27 |
| 100% | 92.82±0.35 | 76.67±0.45 | 21.63±0.24 |

## 5 Conclusion

This work is the first to identify community bias in medical VLP stemming from diverse language communities and illustrates its negative impact on various downstream tasks. We present Med-UniC, a novel cross-lingual medical VLP framework, along with CTR, intended to unify cross-lingual text representations and mitigate language-driven community bias. This bias impacts both text and visual modalities, thereby affecting performance across vision-language and uni-modal visual tasks. The superior performance of Med-UniC across various tasks and data ratios underscores its efficiency and effectiveness. Through comprehensive ablation studies, we show that CTR significantly enhances performance in both vision-language and uni-modal visual tasks by effectively reducing community bias. This study provides a robust cross-lingual medical VLP framework and emphasizes the importance of inclusivity beyond English-speaking communities.

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
