# Med-UniC: Unifying Cross-Lingual Medical Vision-Language Pre-Training by Diminishing Bias (Supplementary Material)

Zhongwei Wan[1]*,    Che Liu[2]*,    Mi Zhang[1]†,    Jie Fu[5]†,    Benyou Wang[4],    Sibo Cheng[2],
Lei Ma[3] ,    César Quilodrán-Casas[2],    Rossella Arcucci[2]

[1]The Ohio State University    [2]Imperial College London
[3]Peking University    [4]The Chinese University of Hong Kong, Shenzhen
[5]The Hong Kong University of Science and Technology

{wan.512, mizhang.1}@osu.edu, lei.ma@pku.edu.cn, wangbenyou@cuhk.edu.cn
{che.liu21, sibo.cheng, c.quilodran, r.arcucci}@imperial.ac.uk, jiefu@ust.hk

## 1 Appendix Overview

In the supplementary material section, we first discuss the social impact and limitations of Med-UniC framework. Then, we introduce the details of the pre-training stage, including cross-lingual MLM pre-training and VLP for Med-UniC. Then we present the detailed configurations of several downstream tasks. Next, we conduct some additional experimental analysis.

## 2 Discussion

**Social impact** Our Med-UniC framework presents an innovative solution for unifying CXR images and corresponding medical reports from diverse communities that use different languages, thereby addressing the data shortage in the medical VLP domain. Further investigations reveal that existing VLP models exhibit biased performance across various linguistic communities, even when pre-trained on cross-lingual data. However, Med-UniC effectively minimizes this bias, enhancing performance not only in cross-lingual tasks but also in uni-modal tasks. In essence, our work offers more than just a strategy for integrating data from different sources; it shines a light on the significant issue of community bias in cross-lingual VLP, calling for more fair and equitable practices in this field.

**Limitation** Our research primarily concentrates on cross-lingual medical VLP, limiting the number of languages, medical images, and reports included due to the current lack of public datasets. We've conducted comprehensive experiments involving uni-modal visual and vision-language tasks, but have not ventured into uni-modal language tasks like report generation. Looking ahead, we aim to expand our work to include more languages in the medical VLP and take on more challenging tasks, such as zero-shot guided segmentation or object detection. This area of our work, thus, acknowledges room for further development and exploration.

**Future work** Exploring these methods on diverse modality medical data such as electrocardiograms paired with clinical monitor records is an interesting future direction [? ]. Furthermore, the alignment of different modality data can be viewed as a data fusion task, a commonly addressed issue in physics [? ? ? ], or recommendation system [? ] .

---

* Equal Contribution.
† Corresponding Authors

37th Conference on Neural Information Processing Systems (NeurIPS 2023).

# 3 Implementation details for Pre-Training

## 3.1 Implementation for Cross-lingual MLM

**Medical Report Pre-processing** We prepare the cross-lingual pre-training corpus from the medical reports of MIMIC (En) [**?** ] and PadChest (Sp) [**?** ]. Specifically, we concatenate the findings and impression to form English medical reports for MIMIC. As for PadChest, we treat their radiology reports as Spanish reports. Examples of these reports can be seen in Fig 1 and Fig 2. To create the MLM corpus, we combine all the reports and shuffle them at the report-level.

**MLM Pre-training Setup** To obtain cross-lingual medical LM, we train the cross-linguistic encoder with the combined corpus using an Adam optimizer and adopt a linear warm-up scheduler. Specifically, the learning rate is 5e-4, and the learning rate linearly increases from 0 to the peak value with a linear warm-up. Additionally, the max length for input tokens is 256. The MLM process is conducted on 8 V100 GPUs and pre-trained within 15 epochs. To save GPU memory and speed up training, we adopt automatic mixed-precision FP16. The details of cross-lingual MLM are shown in Tab 1.

Table 1: Hyper-parameters of Cross-lingual MLM

| Hyperparameters | |
|---|---|
| Training epochs | 15 |
| Total Batch size | 1024 |
| Number of GPUs | 8 |
| Gradient Accumulation | 16 |
| Maximum Sequence Length | 256 |
| Learning Rate | 5e-4 |
| Learning Rate Optimizer | Adam |
| Schedule | Linear Warm-up |
| Warm-up Proportion | 10% |
| Adam Epsilon | 1e-8 |
| Masked Rate | 0.15 |
| FP16 | True |

Table 2: Hyper-parameters of VLP

| Hyperparameters | |
|---|---|
| Pre-training epochs | 50 |
| Batch size per GPU | 128 |
| Number of GPUs | 16 |
| Gradient Accumulation | 2 |
| Maximum Sequence Length | 256 |
| Learning Rate | 4e-5 |
| Learning Rate Optimizer | AdamW |
| schedule | CosineAnnealing |
| Weight Decay | 5e-2 |
| FP16 | True |
| Frozen Linguistic Encoder Layers | 9 |
| $\lambda$ | 5.1e-3 |
| VL Alignment Dimension | 512 |
| CTR Embedding Dimension | 1024 |

## 3.2 Implementation for Vision-language Pre-training

**Model Architecture** Following the same framework as CLIP [**?** ], we utilize ResNet-50 [**?** ], ViT-B/16 and ViT-L/32 [**?** ] as our visual encoder, and we further pre-train CXR-BERT [**?** ] via cross-lingual masked language modelling to obtain the linguistic encoder. Moreover, the input resolution of visual encoder is $256 \times 256$, and the input token length of the linguistic encoder is 256. The final image and text features are projected to the same dimension, which is 512 as [**?** ], followed by batch normalization before interaction. The dimension of cross-lingual text alignment regularization (CTR) is set to 1024. More senstivity analysis can be found in Sec 5.5.

**Image Data Pre-processing** The original CXR images from the MIMIC-CXR and PadChest datasets [**?** **?** ] are resized to $256 \times 256$ and randomly cropped to $224 \times 224$, following the procedure in [**?** **?** **?** ]. All images are normalized to the range $[0, 1]$. For data augmentation during pre-training, we apply horizontal flip, random rotation in the range $[0°, 180°]$, and auto contrast using the PyTorch vision library[1]. The English and Spanish CXR image examples and its corresponding reports are depicted in Fig 1 and Fig 2, respectively.

**Vision-language Pre-training Setup** The detailed hyperparameters of vision-language pre-training for Med-UniC are shown in Tab 2. We use a cosine annealing scheduler for the adjustment of the learning rate. The pre-training step is conducted by V100 GPUs. To save GPU memory and speed up training, we also adopt automatic mixed-precision FP16.

# 4 Configurations for downstream Tasks

This section provides a detailed introduction to all downstream tasks and presents the data split information for each task. The data split details are presented in Tab 3.

---

[1]https://pytorch.org/vision/stable/transforms.htmls

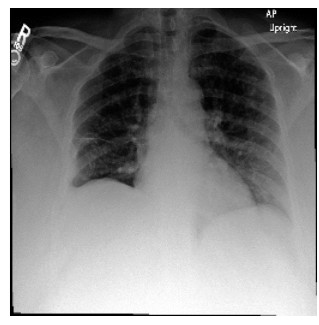

(a) CXR image example 1 from MIMIC dataset

**Subject ID**: 10000935
**Study ID**: 56164612
**Report:**
Lung volumes are low The heart size is normal.
The mediastinal and hilar contours are unremarkable.
New nodular opacities are clustered within the left upper lobe, and to a lesser extent, within the right upper lobe.
……
Findings are compatible with metastases, as was noted in the lung bases on the subsequent CT of the abdomen and pelvis performed later the same day.

(b) CXR report example 1 from MIMIC dataset

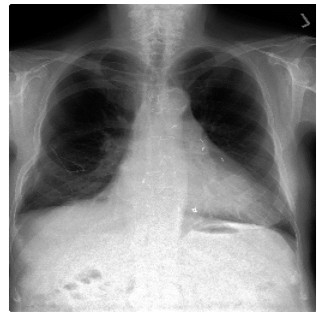

(c) CXR image example 2 from MIMIC dataset

**Subject ID:** 10002430
**Study ID:** 58911492
**Report:**
The lung volumes are normal.
Mild cardiomegaly which is stable.
Normal hilar and mediastinal structures.
……
Status post CABG with aligned median sternotomy wires and normal location of surgical clips.
Status post right lung surgery with surgical material seen.
Mild cardiomegaly No evidence of pneumonia

(d) CXR report example 2 from MIMIC dataset

Figure 1: CXR dataset examples from MIMIC-CXR[**?** ].

Table 3: Data Split Details, '/' indicates that no training/validation data is required in the zero-shot classification task.

| Task | Dataset | Split | Train | Valid | Test |
|---|---|---|---|---|---|
| Linear Classification | CheXpert [**?** ] | [**?** ] | 186,027 | 5,000 | 202 |
| | RSNA [**?** ] | [**? ?** ] | 16,010 | 5,337 | 5,337 |
| | COVIDx [**?** ] | [**? ?** ] | 23988 | 5998 | 400 |
| Semantic Segmentation | RSNA [**?** ] | [**? ?** ] | 16,010 | 5,337 | 5,337 |
| | SIIM [**?** ] | [**? ?** ] | 8,433 | 1,807 | 1,807 |
| Object Detection | RSNA [**?** ] | [**? ?** ] | 16,010 | 5,337 | 5,337 |
| | Object-CXR [**?** ] | [**?** ] | 6,400 | 1,600 | 1,000 |
| Zero-shot Classification | CXP500 [**?** ] | [**?** ] | / | / | 500 |
| | PDC [**?** ] | [**?** ] | / | / | 1000 |

**Medical Image Linear Classification** We explain the setting of linear classification tasks, including CheXpert [**?** ], RSNA [**?** ], and COVIDx [**?** ]. We utilize ResNet-50, ViT-B/16 and ViT-L/32 as our visual backbone and fine-tune the linear layer with 50 epochs using early stop, with the same learning rate of 5e-4 and the default batch size is 8. We leverage the AdamW optimizer to schedule the learning rate, with the $\beta_1$ of 0.9, the $\beta_2$ of 0.999, and the weight decay rate of 1e-6. All the linear classification tasks are conducted on a V100 GPU with 32GB memory.

**Medical Image Semantic Segmentation** For the segmentation tasks RSNA and SIIM [**?** ], we first adopt the ResNet-50 as the visual backbone and train the segmentation network on a 32G V100 GPU. We leverage early stopping to train the tasks for 50 and 100 epochs. We adapt 5e-4 as the learning rate , 1e-8 as the weight decay and 4 as the default batch size. We also employ AdamW as the optimizer

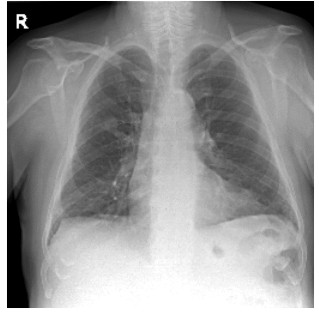

**Study ID:** 216840111366964013217898866992011329134906098
**Report:**
Nodul proyect lsd 1 4 cm contorn parcial bien defin cit tac torac.
Patron intersticial con probabl are panalizacion bibasal probabl
fibrosis pulmon.

(a) CXR image example 1 from PadChest dataset

(b) CXR image example 1 from PadChest dataset

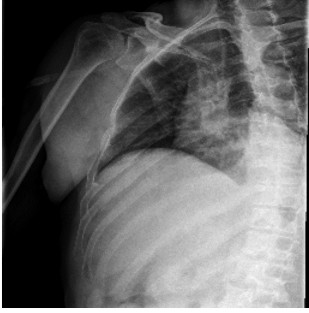

**Study ID:** 216840111366964013515091760022012318080539431
**Report:**
Fractur arcos costal lateral derech cuart quint sext septim.
Condensacion subsegmentari lii descart aspiracion sobreinfeccion
respiratori.
Pinzamient sen costofren izquierd.

(c) CXR image example 2 from PadChest dataset

(d) CXR image example 2 from PadChest dataset

Figure 2: CXR dataset examples from PadChest [**?** ].

and set the $\beta_1$ and $\beta_2$ as 0.9 and 0.999, respectively. For the ViT-B/16 and ViT-L/32 backbones, we set the training configurations following [**? ?** ].

**Medical Image Object Detection** The Object Detection tasks RSNA and Object-CXR [**?** ] are conducted on 1 V100 GPU. We use the grid search to find the optimal batch size as shown in Tab **??**. Specifically, we also adapt AdamW as our optimizer with the learning rate of 5e-4, weight decay of 1e-6, $\beta_1$, $\beta_2$ of 0.9 and 0.999, batch size of 4. The IOU and NMS thresholds are [0.4, 0.45, 0.5, 0.55,0.6, 0.65, 0.7, 0.75] and 0.5, respectively. We use ResNet-50 as the visual encoder of Med-UinC for fair comparisons with other baselines.

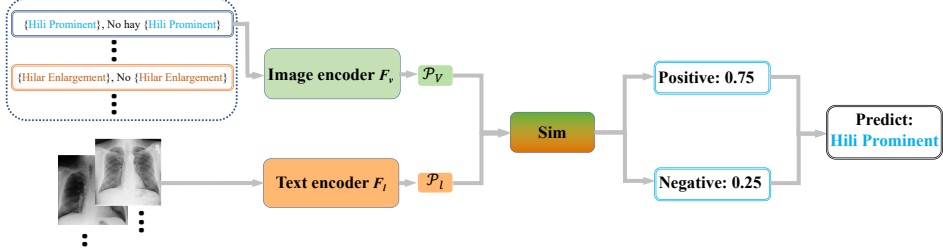

Figure 3: An example of Med-UniC zero-shot pipeline. Blue and orange denote Spanish and English prompts, respectively.

**Zero-shot Task** The original image undergoes a two-step process. Firstly, it is resized to dimensions of $256 \times 256$ and then center cropped to $224 \times 224$. Subsequently, all pixel values are normalized within the range of $[0, 1]$, following [**? ?** ]. The resulting resized image is then passed through an image encoder to generate an image embedding . Concurrently, the prompts are inputted into a text encoder to obtain a text embeddings. To evaluate the classification, we measure the cosine similarity between the image and text embeddings for each prompt associated with a specific class. This is

Table 4: Comparisons of different backbones on medical image segmentation tasks. Best results of each setting are in boldface.

| Backbone | RSNA | | | SIIM | | |
|----------|------|------|------|------|------|------|
| | 1% | 10% | 100% | 1% | 10% | 100% |
| ResNet-50 | 72.6 | 74.4 | 76.7 | 56.7 | 62.2 | 64.4 |
| ViT-B/16 | **75.6** | 76.6 | 77.9 | 62.1 | 67.3 | 71.5 |
| ViT-L/32 | 74.4 | **77.5** | **78.7** | **64.5** | **68.7** | **73.3** |

computed using the equation:

$$\text{sim(img, prompt)} = \mathcal{P}_v(\mathbf{v}_{e,i})^T \mathcal{P}_l(\mathbf{l}_{e,i}) \tag{1}$$

where $\mathbf{v}_{e,i}$ and $\mathbf{l}_{e,i}$ are image and text embeddings from visual and linguistic encoders, respectively. $\mathcal{P}_v$ and $\mathcal{P}_l$ are visual and linguistic projectors. The classification outcome is determined based on the comparison of cosine similarities. If the cosine similarity between the image embedding and the positive prompt (e.g., _disease_) exceeds the cosine similarity between the image embedding and the corresponding negative prompt (e.g., _No disease_), the outcome is considered positive. Conversely, if the reverse is true, the outcome is negative. The pipeline is detailed in Fig 3. To compute the performance metrics, the macro AUC score and F1 score, we calculate these scores for each category individually and then average them across all categories.

# 5 Additional Experimental Analysis

## 5.1 Tokenization Analysis of Cross-lingual Medical LM

**Input Spanish report:**

'infitr alveor ambos camp pulmonar predomini derech bibasal sin grand cambi relacion rx previ 11 03 2017 . derram pleural bilateral . catet venos con extrem proyect vcs .'

**Tokens of CXR-BERT tokenizer:**

['inf', '##itr', 'alve', '##or', 'amb', '##os', 'camp', 'pulmon', '##ar', 'predom', '##ini', 'dere', '##ch', 'bibas', '##al', 'sin', 'grand', 'camb', '##i', 'rel', '##aci', '##on', 'rx', 'prev', '##i', '11', '03', '2017', '.', 'der', '##ram', 'pleural', 'bilateral', '.', 'cat', '##et', 'veno', '##s', 'con', 'extrem', 'pro', '##ye', '##ct', 'vc', '##s', '.']

**Tokens of Cross-lingual medical LM tokenizer:**

['infitr', 'alveor', 'amb', '##os', 'camp', 'pulmonar', 'predomini', 'derech', 'bibasal', 'sin', 'grand', 'cambi', 'relacion', 'rx', 'previ', '11', '03', '2017', '.', 'derram', 'pleural', 'bilateral', '.', 'catet', 'venos', 'con', 'extrem', 'proyect', 'vcs', '.']

Figure 4: Tokenization visualization of one report sample from PadChest dataset. Blue tokens represent tokenized words in the Spanish medical vocabulary of cross-lingual medical LM.

In this section, we visualize the tokenization results of one random selected Spanish report generated by CXR-BERT and Cross-lingual Medical LM' tokenizers, respectively. As shown in Fig 4, compared with CXR-BERT, Cross-lingual Medical LM's tokenizer can handle the Spanish sentence correctly and give a tokenized scheme without loss of Spanish words' semantics. However, CXR-BERT's tokenizer uses tokens from its English medical vocabulary may deteriorate the original Spanish semantics. The visualization results demonstrate the importance of building a cross-lingual lexicon from Spanish medical reports.

## 5.2 Medical Image Segmentation with ViT Backbone

In this section, we adopt the ViT-B/16 and ViT-L/32 as the visual encoder of Med-UniC to compare with ResNet-50 for medical image segmentation tasks on RSNA and SIIM datasets. As shown in Tab 4, surprisingly, the ViT-B/16 backbone has noticeable improvement compared with the ResNet-50 backbone, especially for SIIM. Additionally, the ViT-L/32 backbone shows better results than ViT-B/16 in most settings, indicating that larger visual backbone can further improve the performance of Med-UniC. We attribute this enhancement from the ViT backbone to the global attention capability of the transformer-based visual encoder on segmentation [**?** ].

Table 5: Ablation Study on CTR. Best results of each setting are in boldface, and the '-' denotes mAP values smaller than 1%.

| | COVIDx(ACC) | | | RSNA(Dice) | | | Object CXR(mAP) | | |
|---|---|---|---|---|---|---|---|---|---|
| Backbone | 1% | 10% | 100% | 1% | 10% | 100% | 1% | 10% | 100% |
| $\mathcal{L}_{CTR}$ | **76.5** | **89.0** | **92.8** | **72.6** | **74.4** | **76.7** | **6.6** | **13.3** | **21.6** |
| $\mathcal{L}_{CTR}$ (w/o $\mathcal{L}_{TF}$) | 74.9 | 86.2 | 92.3 | 72.2 | 72.9 | 74.4 | - | 12.3 | 19.9 |
| $\mathcal{L}_{CTR}$ (w/o $\mathcal{L}_{TT}$) | 75.8 | 87.4 | 92.5 | 71.6 | 73.5 | 75.6 | 4.5 | 12.9 | 20.5 |

Table 6: Ablation Study on VLP data, The best results for each setting are highlighted in bold, and the '-' denotes mAP values smaller than 1%. Methods with ⋆ leverage disease-level annotations. [†] denotes that Med-UniC only pre-trained on MIMIC dataset (En). Best results of each setting are in boldface.

| | COVIDx(ACC) | | | RSNA(Dice) | | | Object CXR(mAP) | | |
|---|---|---|---|---|---|---|---|---|---|
| Backbone | 1% | 10% | 100% | 1% | 10% | 100% | 1% | 10% | 100% |
| ConVIRT[? ] | 67.3 | 77.8 | 89.0 | 55.0 | 67.4 | 67.5 | - | 8.6 | 15.9 |
| GLoRIA-MIMIC[? ] | 66.5 | 80.5 | 88.8 | 60.3 | 68.7 | 68.3 | - | 8.9 | 16.6 |
| MGCA⋆ [? ] | 74.5 | 85.2 | 90.3 | 63.0 | 68.3 | 69.8 | - | 12.1 | 19.2 |
| MedKLIP⋆ [? ] | 72.0 | 83.5 | 90.5 | 66.2 | 69.4 | 71.9 | - | 7.1 | 11.6 |
| Med-UniC (En)[†] | 74.7 | 85.5 | 91.3 | 70.4 | 71.2 | 73.5 | 3.4 | 12.5 | 19.5 |
| Med-UniC | **76.5** | **89.0** | **92.8** | **72.6** | **74.4** | **76.7** | **6.6** | **13.3** | **21.6** |

### 5.3 Ablation Study on Cross-lingual Text Alignment Regularization

In this part, we explore the influence of each sub-component belonging to Cross-lingual Text Alignment Regularization $\mathcal{L}_{CTR}$, including text-feature alignment $\mathcal{L}_{TF}$ and text-to-text alignment $\mathcal{L}_{TT}$. As shown in Tab 5, reducing $\mathcal{L}_{TT}$ or $\mathcal{L}_{TF}$ can both lead to performance drop. Specifically, removing $\mathcal{L}_{TF}$ shows worse results than removing $\mathcal{L}_{TT}$ on most of settings, demonstrating the importance of learning cross-lingual invariance and reducing the languages bias through $\mathcal{L}_{TF}$.

### 5.4 Med-UniC Pre-training on Uni-lingual data

On the basis of Med-UniC paradigm, we further study the performance of only using the uni-lingual MIMIC dataset for pre-training. The ablation results are shown in Tab 6, which reveals that although using uni-lingual data to pre-train Med-UniC causes performance drop, compared to other baselines, our framework can also bring benefits. We attribute this to the self-supervised vision alignment (SSV) to learn more exhaustive visual representation and text-to-text alignment to keep in-modal consistency.

Table 7: Dimension Analysis of Projector $\mathcal{P}_d$. Best results of each setting are in boldface.

| | COVIDx(ACC) | | | RSNA(Dice) | | | Object CXR(mAP) | | |
|---|---|---|---|---|---|---|---|---|---|
| Dimension | 1% | 10% | 100% | 1% | 10% | 100% | 1% | 10% | 100% |
| 512 | 75.5 | 87.9 | 92.2 | 72.4 | 73.8 | 74.5 | 3.8 | 12.9 | 20.8 |
| 1024 | **76.5** | **89.0** | **92.8** | 72.6 | **74.4** | **76.7** | **6.6** | **13.3** | 21.6 |
| 2048 | 75.9 | 88.7 | 92.5 | **73.1** | 74.2 | 76.5 | 4.6 | 13.1 | **22.5** |

### 5.5 Embedding Dimension Analysis of Text Alignment Projector $\mathcal{P}_d$

In this section, we investigate the impact of output dimension $D'$ from text alignment projector $\mathcal{P}_d$, since $D'$ determines the size of cross-correlation matrix $\tilde{\mathbf{Z}}_e$ for text-feature alignment. We experiment with linear classification, segmentation and object detection, respectively. Tab 7 shows the corresponding results with different dimensions. When the dimension size is 1024, it can achieve

better results on most of the experimental settings. Hence, we take $D' = 1024$ as our default setting during the pre-training.