# OpenReview forum: "Med-UniC: Unifying Cross-Lingual Medical Vision-Language Pre-Training by Diminishing Bias"
_NeurIPS.cc/2023/Conference — NeurIPS 2023 poster_

### Official Review · Reviewer_uyBA · 2023-07-05

**Soundness:** 3 good
**Presentation:** 2 fair
**Contribution:** 2 fair
**Rating:** 5
**Confidence:** 3

**Summary:**

This paper proposes a vision language pretraining method that focusing on tackling the bias caused by different languages. The Text Alignment Regularization (CTR) is proposed to unify cross-lingual semantic representations of medical reports. The experiments show that the proposed CTR can effectively eliminate the bias between English and Spanish medical report.

**Strengths:**

1. This paper address an important and interesting issue of the bias caused by different languages in medical visual language pretraining tasks.
2. The experiments show the  performance superiority of the proposed method on both normal medical recognition tasks and cross-linguistic tasks.

**Weaknesses:**

1. The design of the proposed visual language model is closed to existing MLM and CLIP based VLP methods with a incremental improvement of CTR. More analysis is needed for the difference between existing methods and existing VLP methods.
2. Seems only zero-shot classification tasks are evaluated under the cross-linguistic setting. Is it possible to evaluate other downstream task under this setting？
3. Can you provide more analysis on why the proposed method outperform other medical VLP methods? It seems like the main difference of this method (CTR) does not really have a strong positive impact on single language recognition tasks.

**Questions:**

Please refer to the concerns in the weakness section.

**Limitations:**

The authors have adequately addressed the limitations

---

> ### Author Rebuttal · Authors · 2023-08-05
>
> ## Response to Reviewer uyBA
>
> ### 1. Response for Weakness 1:
> > The design of the proposed visual language model is closed to existing MLM and CLIP based VLP methods with a incremental improvement of CTR. More analysis is needed for the difference between existing methods and existing VLP methods.
>
> We deeply appreciate your insightful critique of our research. We would like to highlight the overall novelty of our proposed approach from the following three aspects compared with existing VLP methods:
>
> - Our proposal to introduce Med-UniC marks a pioneering step in investigating $\textbf{the bias influenced by language differences}$ within the field of medical VLP. This innovative approach $\textbf{is acknowledged by reviewers}$  `YFc1` and `ZNkV`.
>
> - We have developed a novel CTR loss strategy aimed at improving the performance of VLP across various downstream tasks, by reducing this bias. To the best of our knowledge, Med-UniC $\textbf{is the first initiative to identify and mitigate}$ language-driven bias in medical VLP through cross-lingual text alignment regularisation (CTR). This innovative approach to CTR design is appreciated by reviewers `ZNkV` and `YFc1`.
>
> - Mostly SOTA methods do not directly impose constraints on text embeddings, which consequently limits their ability to extract high-level semantics from text. For instance, MRM [1], despite utilising Masked Language Modeling (MLM) as a pre-training objective for the text, is mainly focused on reconstructing masked tokens rather than learning the overall embeddings of entire sentences [2]. As a solution, we propose the use of CTR to improve the learning of comprehensive sentence embeddings through contrastive learning for each text sample via $L_{TT}$. Additionally, CTR aids in disentangling the text's latent space through $L_{TF}$ to maximise the information on each latent dimension [3].
>
> References
>
> [1] Zhou, Hong-Yu, et al. "Advancing Radiograph Representation Learning with Masked Record Modeling." The Eleventh International Conference on Learning Representations. 2023.
>
> [2] Neelakantan, Arvind (OpenAI), et al. "Text and code embeddings by contrastive pre-training." arXiv preprint arXiv:2201.10005 (2022).
>
> [3] Zbontar, Jure, et al. "Barlow twins: Self-supervised learning via redundancy reduction." International Conference on Machine Learning. PMLR, 2021.
>
> ### 2.  Response for Weakness 2:
> > Seems only zero-shot classification tasks are evaluated under the cross-linguistic setting. Is it possible to evaluate other downstream task under this setting？
>
> We greatly value your constructive feedback on our research. It's important to note that the other downstream tasks we have investigated - specifically, image classification, segmentation, and detection  $\textbf{solely rely on images as input}$ , hence not necessitating any cross-lingual configurations.
>
> ### 3.  Response for Weakness 3:
>
> > Can you provide more analysis on why the proposed method outperform other medical VLP methods? It seems like the main difference of this method (CTR) does not really have a strong positive impact on single language recognition tasks.
>
> We would like to express our sincere gratitude for your thoughtful feedback regarding our research. As demonstrated in $\textbf{Appendix Tab 6}$ , Med-UniC, equipped with the CTR loss and solely pre-trained on an English dataset, outperforms all baselines across three separate vision tasks. This is despite our configuration only utilising uni-lingual Bert and thus, $\textbf{not implementing MLM loss}$ in this pretraining setting. This performance boost may result from the disentanglement effect that the CTR loss has on the text's latent space. This effect, in turn, allows the model to become more adept at extracting high-level semantics and representing features with greater effectiveness. Furthermore, we compare the backbone with CTR and without CTR in $\textbf{Tab A of the attached PDF}$ in the author rebuttal. As the table shows, the CTR loss brings substantial improvement.

---

> ### Author Response · Authors · 2023-08-17
>
> Dear Reviewer, We are deeply grateful for the attention and care you've given to our work. Understanding the importance of thorough feedback, we're here to address any queries or points of ambiguity regarding our response. Please feel free to reach out with any further questions.

---

### Official Review · Reviewer_NLjC · 2023-07-06

**Soundness:** 2 fair
**Presentation:** 3 good
**Contribution:** 2 fair
**Rating:** 6
**Confidence:** 4

**Summary:**

This paper presents a unified framework for Cross-Lingual Medical Vision-Language Pre-Training (Med-UniC), integrating multimodal medical data from different languages (e.g., English and Spanish). A Cross-lingual Text Alignment Regularization (CTR) is proposed to explicitly unify cross-lingual semantic representations of medical reports originating from diverse language communities. It reaches superior performance across 5 medical image tasks and 10 datasets encompassing over 30 diseases.

**Strengths:**

1. Practical and interesting problem setting, which attempts to unifying the Medical Vision-Language Pre-Training for multiple languages.
2. The method is straight-forward and simple, making the paper easy to understand.
3. The Cross-lingual Alignment loss seems reasonable.

**Weaknesses:**

1. The experimental setting is unclear. For example, the implementation and the training data of the SOTA methods are not introduced.
2. The ablation study seems incomplete, only four settings are shown. It is hard to tell how much gain actually comes from the key design of the paper, i.e., the CTR loss. The biggest gain seems to come from the component "MLM", which is not introduced in the paper. I would also suggest the authors to provide ablation showing how much gain is from the increase in data quantity.


**Questions:**

1. How are the SOTA methods implemented? What data set are they trained on? I wonder whether the comparison is fair if the SOTA methods and the proposed method are trained with different set of data. If the SOTA methods are only trained with one language, I wonder how much gain comes from the additional data.
2. Table 1. Why for CXP500, most SOTA methods show a decrease in performance when the prompts transition from English to Spanish on CXP500, but an increase on PDC? The proposed method shows a consistent decrease in performance when the prompts transition from English to Spanish on both data sets.
3. What does the learning objective “MLM” mean in Table 4?

**Limitations:**

No discussion on limitations

---

> ### Author Rebuttal · Authors · 2023-08-05
>
> ## Response to Reviewer NLjC:
> ### 1.  Response for Weakness 1:
>
> We sincerely appreciate your insightful feedback about our research. Concerning the results from other SOTA methods, these $\textbf{were all pre-trained on MIMIC-CXR}$, the English dataset, with the exception of GLoRIA, which was pre-trained using in-house data. Results specifically pertaining to GLoRIA pre-trained on MIMIC-CXR are highlighted as GLoRIA-MIMIC. We directly adopted these results $\textbf{from their original publications}$. Where experimental results were not documented in the original papers, we obtained their official checkpoints and used identical settings to those in our work. Thorough details of the implementation for pretrain and all downstream tasks can be found in appendix Sec B and C. The clear experimental setting with comprehensive experiments $\textbf{are also acknowledged by}$ reviewer `ZNkV`.
>
> ### 2. Response for Weakness 2:
>
> > The ablation study seems incomplete, only four settings are shown.
>
> Please refer to $\textbf{Tab 1 and 6 of the main paper}$ , and $\textbf{Tab 4, 5, 6 , and 7 in the appendix}$. We have done additional ablation study for Med-UniC.
>
> - 1. We ablate three different visual backbones on the medical image classification task in Tab 1 of main paper.
> - 2. We ablate the impact on performance from the number of frozen layers in language model in Tab 6 of the main paper.
> - 3. We also further ablate three different visual backbones  on the medical image segmentation task in Tab 4 in appendix.
> - 4. We ablate  each sub-component's influence belonging to Cross-lingual Text Alignment Regularization (CTR), including the impact of text-feature alignment $L_{TF}$ and text-to-text alignment $L_{TT}$ in Tab 5 in appendix.
> - 5. We study the performance of only using the English dataset, MIMIC-CXR for pre-training without MLM but with CTR in Tab 6 of appendix.
> - 6. We analyse the dimension of the text alignment projector $\mathcal{P}_{d}$ in Tab 7 in the appendix.
>
> Hope those further ablation studies resolve your confusion.
>
> > It is hard to tell how much gain actually comes from the key design of the paper, i.e., the CTR loss.
>
> From $\textbf{Tab 4}$ of main paper and $\textbf{Tab A  in the attached PDF}$, we were able to clearly observe that CTR is a crucial component of our model. Compared to the full version of the model, without CTR causes more performance drop than without MLM (Masked language modelling). Besides, the superior performance of the full model proves that the CTR component is complementary to the MLM strategy, because we leverage MLM to initialise Med-UniC with the ability to process different languages and learn fundamental cross-lingual syntactic and semantic knowledge. But we find the model bias stemming from the language model (LM) pre-trained on predominantly English corpora. To eliminate language bias , we further optimise our model with CTR loss. Therefore, in Tab 4 of the main paper, the comparison of two rows (w/o CTR and with CTR) that the CTR loss leads to excellent performance gains.
>
> > The biggest gain seems to come from the component "MLM", which is not introduced in the paper.
>
> Please refer to $\textbf{Fig 1 or  Sec 3.2}$ in our paper, where we have described the definition of MLM (Masked language modelling).
>
>
> > I would also suggest the authors to provide ablation showing how much gain is from the increase in data quantity.
>
> Please refer to $\textbf{Appendix Sec D4: Med-UniC Pre-training on Uni-lingual data}$,
> The ablation results are shown in Tab 6 in D4, which reveals that although using uni-lingual data to pre-train Med-UniC causes performance drop, compared to other baselines,our framework can also bring benefits. We attribute this to the text-to-text alignment to keep in-modal consistency and better extract text semantics.
>
> ### 3.  Response for Question 1:
>
> To fairly compare Med-UniC with other baselines without additional Spanish dataset, we have conducted an ablation study detailed in $\textbf{Tab 6 in the Appendix}$. For this experiment, Med-UniC was solely pre-trained on MIMIC-CXR, and our model continues to significantly outperform all baselines across three disparate vision tasks. This result corroborates that the effectiveness of Med-Unic does not solely depend on the additional data, but also capitalises on the innovative CTR loss to further extract the nuanced, high-level semantics of the text.
>
> ### 4.  Response for Question 2:
>
> We are deeply appreciative of your constructive critique concerning our research. Initially, it is significant to note that despite a minor decline observed in the transition from English prompts to Spanish in the zero-shot classification task, our approach still holds superiority over all existing baselines. Additionally, referring to Table 1 in the primary manuscript, the F1 scores for most baselines consistently exhibit a downturn from English to Spanish on the PDC task. Even though there is a marginal ascent observed in the AUC scores from English prompts to Spanish, the values predominantly hover around 50. This statistic suggests a rather poor performance [1]. Hence, the minor increment observed could potentially be attributed to random fluctuations stemming from mis-tokenization, as elaborated in $\textbf{Appendix Sec D.1}$.
>
> Reference:
>
> [1] de Hond, et al."Interpreting area under the receiver operating characteristic curve." The Lancet Digital Health 4.12 (2022): e853-e855.
>
>
> ### 5. Response for Question 3:
>
> > What does the learning objective “MLM” mean in Tab 4?
>
> Thanks for your comments, MLM means Masked Language Modeling [1], and it is a type of language modelling task aimed to train a model to predict masked or hidden words in a sentence or text given the surrounding context to learn the semantic information.
>
> Reference:
>
> [1] Kenton, et al. "BERT: Pre-training of Deep Bidirectional Transformers for Language Understanding." Proceedings of NAACL-HLT. 2019.

---

> > ### Comment · Reviewer_NLjC · 2023-08-17
> >
> > Thanks for the detailed response which has cleared most of my concerns. I have raised my rating.

---

> > > ### Author Response · Authors · 2023-08-17
> > >
> > > We sincerely appreciate you taking the time to read our rebuttal and update the rating. Your feedback is deeply valued and we are always here to assist further.

---

> ### Author Response · Authors · 2023-08-17
>
> Dear Reviewer, Your thoughtful review of our work is profoundly appreciated. We recognize the dedication it takes to provide such feedback. If there are areas in our response that need further clarification, or if additional questions arise, we stand ready to engage and offer any necessary assistance.

---

### Official Review · Reviewer_ZNkV · 2023-07-07

**Soundness:** 3 good
**Presentation:** 3 good
**Contribution:** 3 good
**Rating:** 7
**Confidence:** 3

**Summary:**

One common challenge in performing medical vision-language pre-training (VLP) is data scarcity, especially in languages other than English. This challenge can be addressed by combining datasets from various languages to train language-agnostic models, but the authors empirically show that each language community (especially non-English ones) induces distinct linguistic biases in their data, even in language-agnostic models. The authors therefore introduce Med-UniC, which leverages cross-lingual text alignment regularization (CTR) (experimented with English and Spanish) to mitigate these biases and achieve SOTA results using chest X-ray scans and reports on many uni-modal visual tasks.

**Strengths:**

•	Appears to be novel, the idea of cross-lingual text alignment regularization (CTR) seems to effectively address a significant per-language bias problem in multilingual models based on self-vision, vision-language, and cross-lingual alignment strategies.
•	Clear architecture explanation with hyperparameters with comprehensive experiments.
•	Strong results on all experiments.
•	Generally well-written


**Weaknesses:**

The bias analysis section seems brief given how much attention it was given in the abstract/intro. The authors state that more analysis is in the appendix, but I would have wanted to see more in the main paper – I’m curious whether they attempted to identify the sources of the bias, or it would have been neat to see some samples with similar (language-agnostic) content but different embeddings due to this bias.

**Questions:**

see above.

---

> ### Author Rebuttal · Authors · 2023-08-05
>
> ## Response to Reviewer ZNkV
> ### 1. Response for Weaknesses:
> > The bias analysis section seems brief given how much attention it was given in the abstract/intro. The authors state that more analysis is in the appendix, but I would have wanted to see more in the main paper .
>
> Thanks for your comment on our paper structure. We will modify the paper layout in the camera-ready version, and move the Sec D1, D3, D4 in the Appendix to the main paper to further explain the bias.
>
>
> > I’m curious whether they attempted to identify the sources of the bias, or it would have been neat to see some samples with similar (language-agnostic) content but different embeddings due to this bias.
>
> To delve deeper into the origin of the bias, we randomly selected 20 reports from the English dataset, translated them into Spanish by native Spanish speakers, and created 20 English-Spanish text pairs which share the same semantic meaning. We computed the correlation coefficient of their embeddings, derived from the uni-lingual Bert, prior to our implementation of cross-lingual MLM pre-training. The correlation coefficient for each sample is depicted in $\textbf{Fig B (left)}$ in the $\textbf{attached PDF of authors rebuttal}$. It's noteworthy that the correlation coefficient for each English-Spanish pair (highlighted as the diagonal elements) does not approach 1.0. For enhanced visual clarity, we have plotted a histogram of the diagonal elements from the correlation coefficient matrix in $\textbf{Fig B (right)}$ in the attached PDF of authors rebuttal, all of which fall below 0.65. This suggests that the uni-lingual Bert perceives the two language versions of the same report, despite having identical semantic meaning, as distinct texts.
>
> Moreover, following the 3D T-SNE method we used in $\textbf{Fig A}$ in our main paper, we visualise the embeddings of Spanish and English reports generated from Uni-lingual Bert (CXR-Bert) in the $\textbf{Fig A a1}$ and $\textbf{Fig A a2}$ in the $\textbf{attached PDF of authors rebuttal}$. From the $\textbf{Fig A}$ in attached PDF, we can clearly observe that the center distance between two language cluster (language bias) is larger than Cross-lingual MLM and Med-UniC (e.g.,  $\textbf{5.22}$(Uni-lingual bert) → $\textbf{2.38}$(Cross-lingual Bert) → $\textbf{0.31}$(Med-UniC)), demonstrating that language models pre-trained predominately on the English corpus produce obvious bias for other languages.

---

> ### Author Response · Authors · 2023-08-17
>
> Dear Reviewer, We deeply appreciate the time and effort you've dedicated to reviewing our work. Your insights are invaluable to us. If any aspect of our response remains unclear or if there are further questions you'd like to discuss, please know that we are more than willing to assist and engage in further dialogue.

---

### Official Review · Reviewer_YFc1 · 2023-07-07

**Soundness:** 3 good
**Presentation:** 3 good
**Contribution:** 3 good
**Rating:** 6
**Confidence:** 4

**Summary:**

The paper aims to address community bias caused by having data in multiple languages in medical vision-language pre-training (VLP). Specifically, it introduces the Unifying Cross-Lingual Medical Vision-Language Pre-Training framework to integrate multi-model data from English and Spanish and proposes a Criss-lingual Text Alignment Regularization (CTR) objective to unify cross-lingual semantic representations of medical reports from different languages. The paper shows superior performance compared to other methods on 5 tasks spanning 10 datasets and provides evidence of community bias in cross-lingual VLP through experimental findings.

**Strengths:**

- Mitigating cultural bias in cross-lingual visual-language datasets in English/Spanish is a significant task with potential applications in the clinical field
- The findings are novel. In particular, the separation between the learned representations in Med-Unic w/o CTR and the clear closer alignment in embedding space in Med-Unic with CTR is very interesting as it shows clearly that the representational spaces of the datasets in English and Spanish were separated without CTR and are better integrated now, which shows the alignment power of the CTR objective
- Results - the paper obtains significant improvement over SOA techniques.
- The CTR loss is an innovative method to address community bias in multi-lingual pre-training methodologies in a mathematically interesting way

**Weaknesses:**

- Looking at Figure 4, it is clear that the pre-training methodology proposed by the authors has significant improvements in bringing the learned representations from the two languages closer in the latent space. However, the two representations are still not perfectly aligned in the latent space, which means that the embeddings are still not perfectly integrated. Full integration would show Figure 4a where the two representations are fully overlapped, whereas now there is still separation between the two, with minimal overlap. While this is still a significant result, the model could probably be improved to better integrate and align the datasets.
- I would like to understand if the authors think that with further training, these two representations can be further aligned, or if this is the maximum representational power capacity of the proposed pre-training methodology.

**Questions:**

-  How was the similarity in Figure 2 between medical reports obtained? What does the X-axis represent?

**Limitations:**

Authors have adequately addressed limitations of their work.

---

> ### Author Rebuttal · Authors · 2023-08-04
>
> ## Response to Reviewer YFc1
> ### 1. Response for Weakness 1 and 2:
> > - Looking at Fig 4, it is clear that the pre-training methodology proposed by the authors has significant improvements in bringing the learned representations from the two languages closer in the latent space. However, the two representations are still not perfectly aligned in the latent space, which means that the embeddings are still not perfectly integrated. Full integration would show Fig 4a where the two representations are fully overlapped, whereas now there is still separation between the two, with minimal overlap. While this is still a significant result, the model could probably be improved to better integrate and align the datasets.
> > - I would like to understand if the authors think that with further training, these two representations can be further aligned, or if this is the maximum representational power capacity of the proposed pre-training methodology.
>
>
> We appreciate your invaluable guidance regarding our research. In addressing the query about Fig 4, the sub-optimal fusion of image representations can be attributed to two fundamental factors. Initially, the two image sets, drawn from disparate communities, do not correspond to the same patient, thereby encapsulating distinct semantic interpretations. This factor could inevitably result in less than perfect integration of the representations. Following on from this, the images' origin diversity, covering a broad range from different hospitals, scanning devices, radiologists, and scanning methodologies, leads to what we identify as a `domain gap`. This inherent variability introduces additional challenges to the process of integration. However, the core thrust of our research is geared towards exploring biases introduced by language. Therefore, we do not forecast a fully congruent representation of images from two distinct communities.
>
> ### 2. Response for Question 1:
> > How was the similarity in Figure 2 between medical reports obtained? What does the X-axis represent?
>
> Thanks for your comment. For the similarity matrix shown in Fig 2 in the main paper, we randomly sample 250 Spanish and 250 English medical reports from both datasets. Then we utilise Cross-lingual Medical LM proposed in Sec.3.2 to obtain the [CLS] (a.k.a classification token) embedding of the texts. Therefore, X-axis represents the index of text embeddings, and the first 250 of these are Spanish samples, and the last 250 are English samples.

---

> ### Author Response · Authors · 2023-08-17
>
> Dear Reviewer, we are truly grateful for your thoughtful feedback on our work. Should you have any additional questions regarding our response, please feel free to ask.

---

> > ### Comment · Reviewer_YFc1 · 2023-08-21
> >
> > Thanks to the authors for the response - makes sense that other types of variability will prevent full overlap between the representations. The response answers my questions.

---

### Author Rebuttal · Authors · 2023-08-05

- We add the graphical explaination of the bias in $\textbf{Fig. A}$ and $\textbf{Fig. B}$  for reviewer `ZNkV`.
- We select the results from Tab 5 in the appendix D.2 and Tab 4 in the main paper to construct $\textbf{Tab. A}$ for reviewer `NLjC`. The $\textbf{Tab. A}$ shows the ablation experimental results of CTR loss.

---

### Decision · Program_Chairs · 2023-09-21

**Decision:**

Accept (poster)

**Comment:**

This paper proposes a framework for medical vision-language pretraining for English and Spanish. The problem they attempt to solve is important and interesting, and the framework is sound. Although there are several limitations (as pointed out in the reviews) including clarity of experiments, mismatched focus on bias between abstract/intro and the main paper, these were adequately addressed in the rebuttal. I encourage the authors to revise the paper according to the rebuttal so that we can all better appreciate the important contributions of this paper.